# Investigation of the Antimicrobial Properties of Beetroot–Gelatin Films Containing Silver Particles Obtained via Green Synthesis

**Judita Puišo** [1,*] ✪, **Diana Adliene** [1,*] ✪, **Algimantas Paškevičius** [2] **and Artūras Vailionis** [3]

1 Department of Physics, Kaunas University of Technology, Studentų Str. 50, LT-51368 Kaunas, Lithuania
2 Laboratory of Biodeterioration Research, Institute of Botany, Nature Research Centre, Akademijos Str. 2, LT-08412 Vilnius, Lithuania
3 Geballe Laboratory for Advanced Materials, Stanford University, McCullough Bldg. 227, 476 Lomita Mall, Stanford, CA 94305, USA
* Correspondence: judita.puiso@ktu.lt (J.P.); diana.adliene@ktu.lt (D.A.);
Tel.: +370-610-04-812 (J.P.); +370-612-08-716 (D.A.)

**Abstract:** Silver nanoparticles are produced using various physical, chemical or physicochemical methods. Simple low-cost nontoxic environmentally friendly, or "green", chemistry methods are important, especially for their invasive application in the medicine and food industries. Silver-nanoparticle-enriched biocompatible films were produced at room temperature from fresh beetroot juice, $AgNO_3$ and gelatin–water solution using the photoreduction method. The optical, morphological and structural characteristics of the experimental samples were analyzed using UV-VIS, XRD and SEM techniques. The antimicrobial activity of newly produced films was investigated using the agar diffusion method. The synthesis of nanoparticles was approved their characteristic LSPR peaks in the UV-VIS absorbance spectra. According to the XRD patterns of the films, these nanoparticles were assigned to the cubic phase of metallic Ag. It was found that the antimicrobial activity of the silver nanoparticles in the beetroot–gelatin films might be effective; however, it depends on the silver ion concentrations used for the production of these films and on the medium's pH.

**Keywords:** silver; nanoparticles; gelatin; surface plasmon; beetroot juice; XRD; UV-VIS; antimicrobial activity

## 1. Introduction

Gelatin is a biodegradable, biocompatible, plastic, adhesive and low-cost biopolymer. Gelatin-based systems are applied in the food, packing, pharmaceutical and photography industries. Due to their unique organoleptic properties, gelatin-based materials are also promising materials for tissue engineering, drug delivery, wound dressing and gene therapy [1]. On the other hand, due to its degradation or hydrolysis in solutions, gelatin is an excellent medium for the growth of most bacteria and for microbial proliferation, thus being a reliable tool for the investigation of the antimicrobial features of materials.

Due to the unique properties of silver (Ag) nanoparticles (NPs), they exhibit a surface plasmon resonance (SPR) peak in the wavelength range of 400 nm–800 nm, allowing their application in multiple fields of technique and medicine [2,3]: in SERS (Surface-Enhanced Raman Spectroscopy) [4], in catalysis [5], as additives for the development of antibacterial materials [6] and also in solar energy harvesting [2,4,7].

Silver nanoparticles are produced through various physical, chemical or physicochemical methods as well as through biological methods [4]. Green synthesis is used as an alternative to these conventional methods. The "hot topic" in the biological synthesis of Ag NPs is the utilization of plants [4,8–10], fruits or their extracts. Plants produce needed nutrients through photosynthesis, converting carbon dioxide and water into sugar fuels

using energy from the Sun. In times of rapid photosynthesis, the main product is glucose, but it is usually converted to the larger sugar sucrose. Sugars (also glucose) have also been used for Ag nanoparticle fabrication as capping and reducing agents [11]. Ag nanoparticles have also been synthesized in natural polymeric matrixes as gelatin, agar–agar or chitosan using silver nitrate, sodium citrate, sodium borohydride or glucose [4]. In order to synthesize the NPs, the plant extract is simply mixed with a solution of silver nitrate, water and a capping agent. Plant extracts may act both as reducing agents and as stabilizing agents in the synthesis of the nanoparticles [11]. The complete reaction time takes a few minutes at room temperature [4].

Beetroots are one of the commonly used vegetables in the food industry [12]. Beetroots (*B. vulgaris* L. subsp. *Conditiva*) have very low saturated fat and cholesterol, vitamin A (1%/100 g), vitamin C (8%/100 g) and minerals: Fe (0.8 mg/100 g), Na (78 mg/100 g), Mg (23 mg/100 g), K (325 mg/100 g) and Mn (0.3 mg/100 g). Beetroots contain carbohydrates (9.6 g/100 g), sugars (6.8 g /100 g), dietary fibers (2.8 g/100 g) and phytosterols (25.0 mg/100 g) [13]. Specific interest in beetroot juice has arisen because it is a rich source of a number of polyphenolic compounds. Beetroots also contain a smaller number of carotenoids and ascorbic acid. Beetroots are a potential source of valuable water-soluble nitrogenous pigments called betalains [12,13]. Betalains consist of two main groups (red betacyanins and yellow betaxanthins).

The betalains of beetroots are used to enhance the red color of different products, such as strawberry ice cream, yogurt, sausages, cooked ham, sauces, jams, biscuits, creams and a range of dessert products, in the food industry [14]. Betalains are important as food colorants and can substitute synthetic as well as expensive natural colorants. Besides their chemical stability, a high tinctorial strength and a constancy in appearance within a broad pH range represent important criteria. Colored extracts are preferred over purified colors because the declaration of the former allows for clean labelling. In this respect, betalains deserve intense research as they offer hues and stability characteristics uncommon to anthocyanin sources [13–16].

Betanin is a purplish-red pigment obtained glycoside of betacyanin consisting of a betaine moiety (indole-2-carboxylic acid) that is N-linked to betalamic acid (pyridine dicarboxylic acid) via an acetyl group with an absorbance of light in the range of 450 nm–600 nm, covering most of the green region [12].

Betaxanthins are known as fluorescent pigments and are able to absorb and emit fluorescence within the visible area of the electromagnetic spectrum. The maximum excitation wavelengths for betaxanthins correspond to blue light and are between 320 and 475 nm. The emission maxima correspond to green light (506 and 660 nm) [12]. Natural plant-derived pigments such as betanin [17] are effective in Ag NP synthesis [18–20].

The aim of this work was to produce silver NP enriched antimicrobial recyclable free-standing films for possible food packaging from a mixture of fresh beetroot juice and gelatin–water solution with a certain amount of AgNO$_3$ admixed using the newly proposed green chemistry method.

## 2. Materials and Methods

### 2.1. Materials

Beetroots were purchased from the local grocery shop in Kaunas, Lithuania (Figure 1). Beetroots were precisely washed, peeled and grated. The resulting mixture was filtered through a 8–12 μm black band filter (Filtrak, Thermalbad Wiesenbad, Germany) to obtain beetroot juices. Silver nitrate powder (AgNO$_3$, CAS No 7761-88-8, ACS reagent, ≥99.9%, Sigma–Aldrich, Poznań, Poland) and gelatin from porcine skin, which had powder gel strength of 250 Bloom, (CAS 9000-70-8, Sigma–Aldrich) were used for film formation.

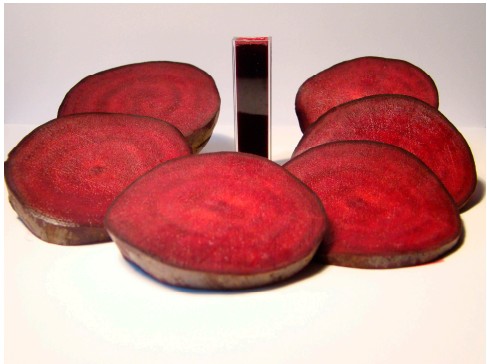

**Figure 1.** A photograph of a sliced beetroot and beetroot juice.

### 2.2. Synthesis of Silver Nanoparticles in Gelatin Matrix

Various amounts (10, 20, 30, 40 or 50 μL) of 1 M AgNO$_3$, which were diluted in distilled water, were admixed to 600 μL of fresh beetroot juice, and the obtained mixture was thoroughly mixed with 20 mL of 10% gelatin–water solution. The estimated beetroot concentration in the mixture was 2.8% *v/v*–2.9% *v/v*. In order to synthesize silver nanoparticles, final mixture was exposed to UV light using 7W black light source characterized by a strong UV emission at 365 nm and a weak blue light emission at 404 nm. Duration of exposure was 15 min. Photograph of cuvettes with corresponding solutions is provided in Figure 2.

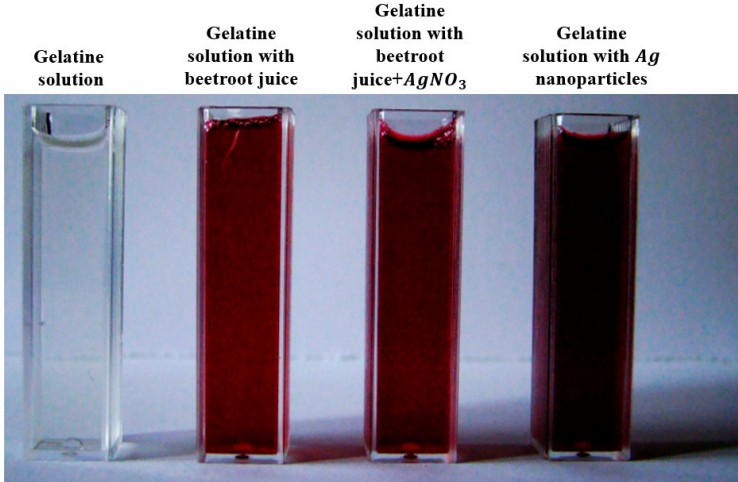

**Figure 2.** A photograph of cuvettes with corresponding solutions. From L to R, gelatin–water solution, beetroot–gelatin–water solution, beetroot–gelatin–silver nitrate solution before UV exposure, and beetroot–gelatin–silver nitrate solution after UV exposure.

Since gelatin–water solution was colorless, it could be seen that the color of beetroot–gelatin solution changed from purple red (contribution of betacyanins present in beetroot) [21–23] to bright red after adding a certain amount of AgNO$_3$ to the previous solution. Color transformation to brownish red after UV exposure of the beetroot–gelatin–AgNO$_3$ mixture indicated formation of colloid with the presence of Ag nanoparticles. Sample color was sensitive to the amount of AgNO$_3$ added to the mixture and to UV irradiation.

### 2.3. Preparation of Films

Weighed amounts of silver colloid (10 g) were casted onto plastic Petri dishes (85 mm) and then dried in open air in a dark box at room temperature (20 ± 2 °C) for 72 h to form 70–80 μm thick free-standing gelatin films (Figure 3).

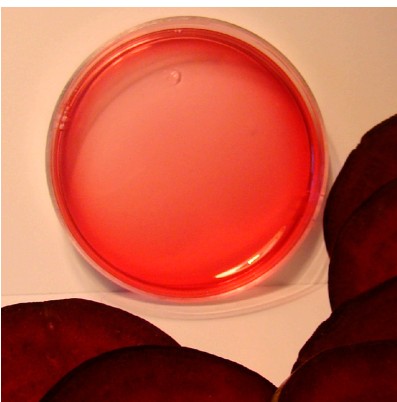

**Figure 3.** An example of beetroot–gelatin free-standing film containing silver nanoparticles.

### 2.4. Characterization of Experimental Samples

The pH of beetroot juice and silver colloid was detected using HI 98130 pH/Conductivity/TDS Tester.

Sugar content in beetroot juice was measured using Abbe Refractometer (AR4 Series).

Optical properties of the experimental samples were investigated using UV-VIS spectrometer Jasco V-650.

The structure of synthesized Ag nanoparticles was investigated using D8 Discover X-ray diffractometer (Bruker AXS GmbH, Karlsruhe, Germany) operating at 40 kV and 40 mA with a Cu K$\alpha$ radiation source ($\lambda$ = 1.5418 Å) and parallel beam geometry with 60 mm Göbel mirror. Diffraction patterns were recorded using a fast-counting LynxEye detector with an opening angle of 2.475° and a slit opening of 6 mm. The peak intensities were scanned over the range of 10–80° (coupled 2θ–θ scans) using a step size of 0.05° and a collection time of 60 s per step.

SEM imaging of samples was performed using scanning electron microscope (JSM-5610 LV) with attached energy dispersive X-ray analyzer (EDX JED-2201; JEOL, Tokyo, Japan).

### 2.5. Microorganisms and Inoculum Preparation

All microorganisms used in this study were isolated from food: bacteria—Escherichia coli BTLB0001 and Pseudomonas aeruginosa BTLB0015, yeasts—*Candida parapsilosis BTLYCP1* and *Yarrowia lipolytica BTLY0054*—and fungi—*Aspergillus terreus BTLF0001*, *A. fumigatus BTLF0002* and *Penicillium expansum BTLF0003*. Bacteria for antimicrobial tests were grown on nutrient agar (Oxoid, Hampshire, UK), and yeasts and fungi were grown on Sabouraud dextrose agar (Oxoid, Hampshire, UK). Inoculum were obtained from overnight bacterial strains on nutrient agar slants at 28 °C. Yeasts were cultured for 3 days, and fungi were cultured for 7 days. The turbidity of the microorganism cell suspensions was measured at 610 nm by spectrophotometer (Thermo Scientific, Waltham, MA, USA). The resulting suspensions of microorganisms were mixed for 15 s with a vortex.

### 2.6. Evaluation of the Antimicrobial Activity of the Beetroot–Gelatin Films Containing Silver Nanoparticles

The activity of gelatin–beetroot films containing silver nanoparticles was screened for antimicrobial activity using the agar diffusion method. For the agar diffusion assay, 100 µL of each suspension of microorganisms was uniformly spread on Mueller–Hinton agar (bacteria) or on Sabouraud dextrose agar (yeasts) in a Petri dish (90 mm). The suspension micromycete spores (one by one mL) were poured into Petri dishes with Sabouraud dextrose agar at 45 °C. The gelatin–beetroot films containing silver nanoparticles (5 × 5 mm) were placed on the surface of each Petri dish after absorption of inoculum by agar. The plates with yeasts and fungi were incubated at 26 ± 1 °C and were placed for 3 days (for fungi) or for 5 days for yeasts, while the plates with bacteria were incubated at 37 °C for 2 days. All tests were triplicated for all strains. Beetroot juice for the synthesis of

silver nanoparticles was used as negative control. The diameters of the inhibition zones after the incubation period were measured in millimeters.

### 2.7. Evaluation of the pH Effect on the Antimicrobial Properties of the Gelatin Films with Silver Nanoparticles

Activities of gelatin–beetroot films containing silver nanoparticles were tested in four pHs, i.e., 3, 4, 7 and 8. The 35% wt. of HCl and 5 M NaOH were used in order to obtain necessary pH values; the amount of base or acid needed in each case was previously determined. For the agar diffusion assay, 100 µL of each suspension of microorganisms was uniformly spread on Mueller–Hinton agar (bacteria) or on Sabouraud dextrose agar (yeast) in a Petri dish (90 mm). The suspensions of spores of micromycetes (one by one mL) were poured into Petri dishes with Sabouraud dextrose agar at 45 °C. The gelatin films with silver nanoparticles (5 × 5 mm) were placed on the surface of each Petri dish after absorption of inoculum by agar. The plates with yeast and fungi were incubated at 26 ± 1 °C for 3 days or for 5 days (for fungi), while plates with bacteria were incubated at 37 °C for 2 days. All tests were triplicated for all strains. Beetroot juice used for the synthesis of silver nanoparticles was used as negative control.

### 3. Results

#### 3.1. Investigation of Physical and Chemical Properties of the Experimental Films

It is known that the beetroot is one of the sweetest vegetables with the highest sugar content. The sugar content of fresh beet root juice was estimated to be 12% *w/v*. Also, fresh beetroot juice is slightly acidic (pH of 5.43).

As it was mentioned in the introduction part of the article, beetroots are a source of water-soluble nitrogenous pigments called betalains consisting of two main groups: betacyanins and betaxanthins [12,13]. The betalain pigments of beetroots are characterized by the overlap of the betanin and betanidin absorbance peaks at 538 nm and 542 nm seen in UV-VIS spectra, respectively [22]. The absorbance peak at 482 nm belongs to betaxanthin [23]. These features create some difficulties when analyzing SPR peaks which are present in the same wavelength interval (400 nm–700 nm) [18,19]. To overcome/reduce this problem the UV-VIS absorbance spectra of water-diluted fresh beetroot juice were analyzed (Figure 4).

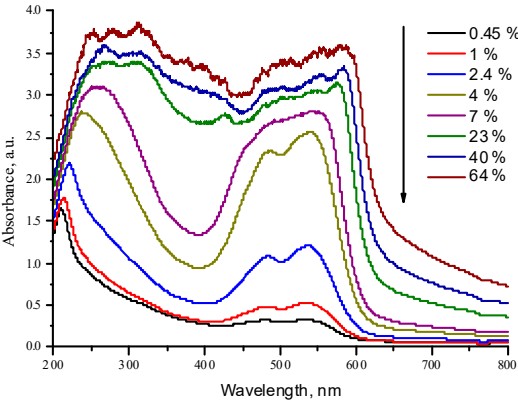

**Figure 4.** UV-VIS spectra of diluted beetroot juice.

The concentrations of the beetroot juice diluted in distilled water varied from 0.45% *v/v* to 64% *v/v*.

The performed analysis revealed that the intensity of the betalain absorbance peaks in the UV-VIS absorbance spectra increased significantly when the beetroot juice concentration in the water solution was ≥4%. Based on the evaluation results, it was suggested that, in order to reduce the influence of betalains on the evaluation of the LSPR peaks of the Ag nanoparticles in the UV-VIS absorbance spectra, the concentration of the beetroot juice in

the final beetroot–gelatin–AgNO₃ solution should be as low as necessary for the production of silver NP containing free-standing gelatin films possessing antibacterial resistance. This correlated well with the produced experimental films, where the concentration of the beetroot juice was 2.8–2.9% depending on the added amount of silver nitrate.

Various amounts (10, 20, 30, 40 or 50 µL) of 1 M AgNO₃, which were diluted in distilled water, were used in the beetroot–gelatin–AgNO₃ solution, which was aimed at producing free-standing films. The results of the investigation of the UV-VIS spectra of the silver-containing beetroot–gelatin free-standing films are presented in Figure 5.

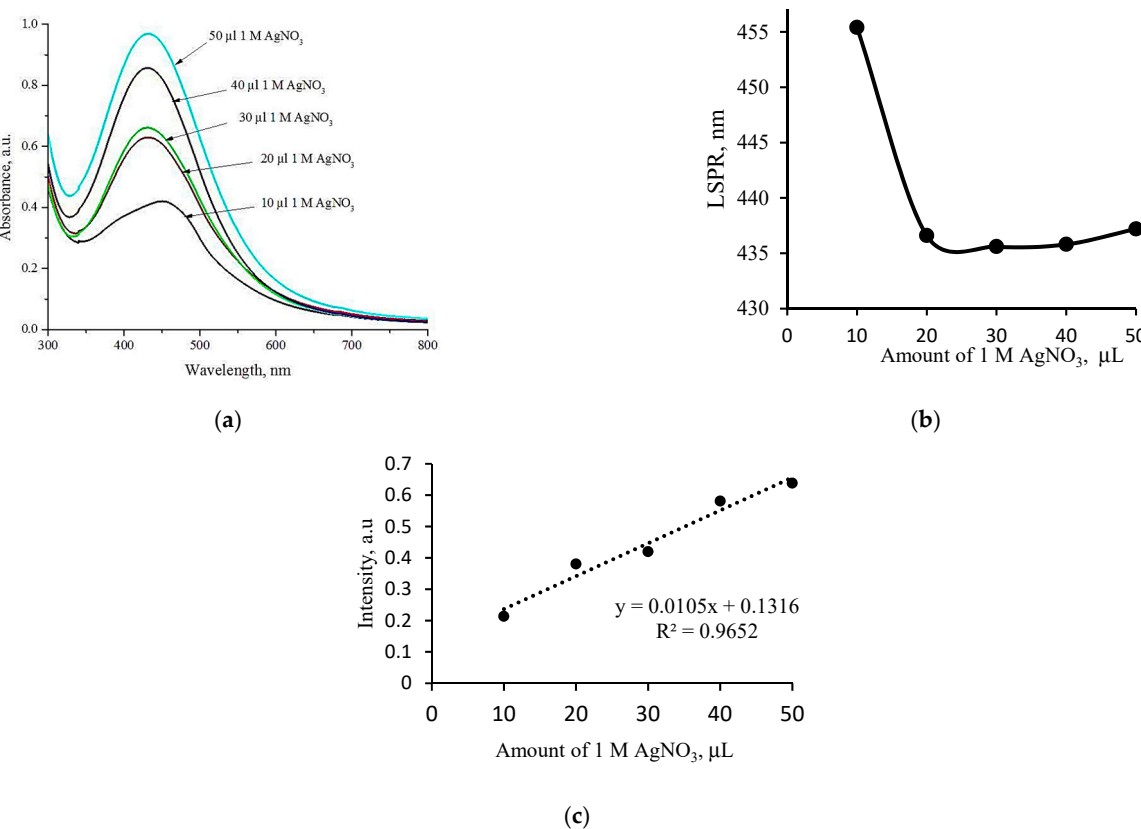

**Figure 5.** Silver-nitrate-concentration-dependent optical properties of silver-enriched beetroot–gelatin nanocomposite films: UV-VIS absorbance spectra (**a**), concentration-dependent LSPR peak position (**b**) and concentration-dependent intensity variations of LSPR peak (**c**).

At low concentrations/amounts of silver ions in the whole mixture, the synthesized silver occurred in the form of seed clusters that were of larger sizes than the nanoparticles. In the clusters, organized seeds indicated only modest and broad absorbance peaks. It seemed that the absorbance peak of betaxanthin at 482 nm from the beetroot juice partly overlapped with the LSPR peak of the silver seeds. The increased concentration of silver ions in the solution contributed to the higher number of synthesized silver nanoderivatives of different sizes which may also stand as separate nanoparticles. Larger NPs may have been produced at higher concentrations. These nanoparticles were characterized by well-pronounced LSPR peaks which became more intensive than the absorbance peak of betaxanthin with increasing silver concentrations. The usage of very small AgNO₃ concentrations in the experimental solutions and the presence of betaxanthin may help to explain the slightly counterintuitive blue shift of the LSPR peak from 455 nm to 435 nm when the AgNO₃ amount in the gel solution was ≤40 µL. At higher silver ion concentrations that may have secured the formation of large enough NPs (up to 100 nm), the opposite (red shift) tendency of the LSRP peak was observed (Figure 5b). The production of larger NPs at higher AgNO₃ concentrations was also supported by the data provided in Figure 5c.

The structure of the synthesized Ag nanoparticles in the beetroot–gelatin films were investigated using the standard XRD powder diffraction procedure. An example of the XRD patterns of the silver-enriched beetroot–gelatin nanocomposite film obtained using 40 μL of AgNO$_3$ is provided in Figure 6a along with the XRD pattern for gelatin (Figure 6b). Actually, only the representative XRD peaks for gelatin at around 6–7 and 20–22 2θ (deg) were observed as was also mentioned in [24]. The typical XRD peaks of the crystalline planes of the face-centered-cubic (fcc) crystalline structure of metallic silver (JCPDS file no. 04-0783) were seen in the XRD pattern of the produced films (Figure 6a): (111) at 38.59°, (200) at 43.2°, (220) at 64.68° and (311) at 77.20°, correspondingly. The obtained results fit well with the results provided by other authors [25,26].

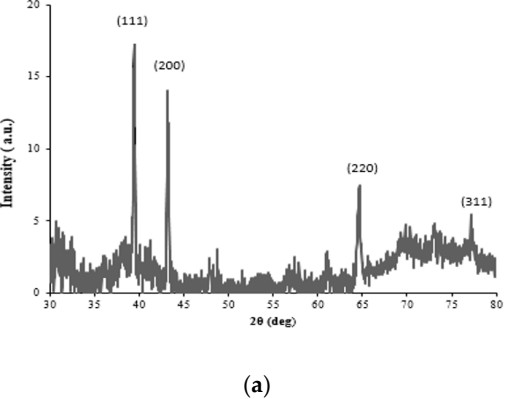

(**a**)

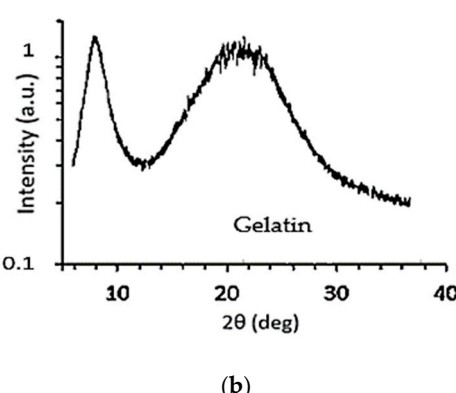

(**b**)

**Figure 6.** X-ray diffraction patterns: full pattern of silver-enriched gelatin–beetroot nanocomposite films (**a**) and pattern of gelatin (**b**).

The morphology of the beetroot–gelatin free-standing films was analyzed using SEM/EDX information as it is indicated in Figure 7. The presence of Ag nanoparticles in the experimental films was clearly seen in the SEM image (Figure 7a). The same was approved through an EDX analysis, which indicated the highest elemental contribution of the Ag NPs along with the other elements observed, such as carbon, oxygen, sodium and chlorine (Figure 7b).

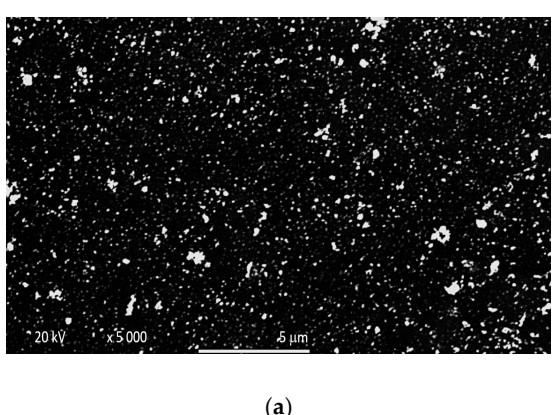

(**a**)

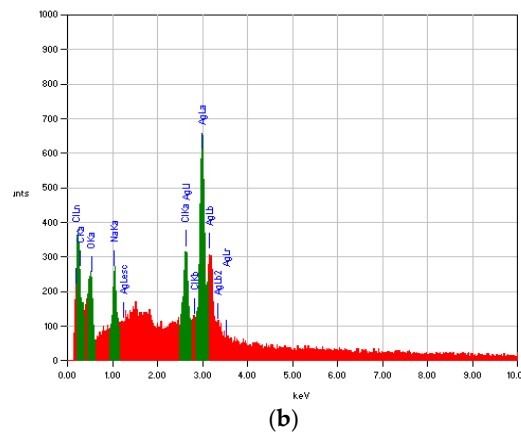

(**b**)

**Figure 7.** SEM image and EDX analysis of the silver-enriched beetroot–gelatin films: SEM image (**a**) and EDX spectra with indicated weighted contribution of chemical elements that were found in the produced films (**b**).

### 3.2. Investigation of the Antimicrobial Properties of the Experimental Films

Gelatin is a biodegradable, biocompatible, plastic, adhesive and low-cost biopolymer. Gelatin-based systems are applied in the food, packing, pharmaceutical and photography

industries. Due to their unique organoleptic properties, gelatin-based materials are promising materials for tissue engineering, drug delivery, wound dressing and gene therapy. On the other hand, gelatin has become an excellent growth medium for most bacteria due to the degradation or hydrolysis of gelatin in solution and due to microbial proliferation [27].

Since it is known that green-synthesized silver nanoparticles are effective in inhibiting the growth of both Gram-positive and Gram-negative bacteria [20], the antimicrobial properties of the beetroot–gelatin films containing silver nanoparticles were tested using agar diffusion methods. The results of the bioassays showed that the beetroot–gelatin films with silver nanoparticles exhibited antimicrobial activity against all the tested bacteria and *C. parapsilosis* (Table 1). The inhibition zones of the beetroot–gelatin films with silver nanoparticles were between 1.0 and 4.0 mm and depended on the microorganism and/or the concentration of the silver nanoparticles in the films. Antifungal activity, however, was not observed in all the fungi cultures or in *Y. lipolytica*.

**Table 1.** Antimicrobial activity of the gelatin films with silver nanoparticles against microorganisms.

| Microorganism | Control (Beetroot Juice) | BuAg10UV | BuAg20UV | BuAg30UV | BuAg40UV | BuAg50UV |
| --- | --- | --- | --- | --- | --- | --- |
| | | Zone Size, mm | | | | |
| *Escherichia coli* | 0 | 2.0 ± 0 | 2.0 ± 0 | 3.0 ± 0 | 2.0 ± 0 | 2.0 ± 0 |
| *Pseudomonas aeruginosa* | 0 | 1.0 ± 0 | 4.0 ± 0 | 1.0 ± 0 | 2.0 ± 0 | 2.0 ± 0 |
| *Candida parapsilosis* | 0 | 2.0 ± 0 | 1.0 ± 0 | 2.0 ± 0 | 2.0 ± 0 | 0 |
| *Yarrowia lipolytica* | 0 | 0 | 0 | 0 | 0 | 0 |
| *Aspergillus terreus* | 0 | 0 | 0 | 0 | 0 | 0 |
| *Aspergillus fumigatus* | 0 | 0 | 0 | 0 | 0 | 0 |
| *Penicillium expansum* | 0 | 0 * | 0 * | 0 * | 0 * | 0 * |

Notice: 0 *—fungi not grown on the gelatin films with silver nanoparticles.

It should be noted that the activity of the microorganisms was pH dependent. It was taken into account that the thermal stability and mechanical properties of the beetroot–gelatin films were dependent on the gelatin's molecular weight distribution, its amino acid composition and the pH of the film-forming solution. The variation of the pH allowed for the release of Ag nanoparticles from the gelatin–beetroot nanocomposite's network. Typically, Young's modulus, the stress break and the thermal stability of the gelatin films significantly decreased when the pH of the gelatin solution was higher than 9.0 or lower than 5.0. The triple-helix structure of the gelatin-based samples was not affected over the pH range of 5.0–9.0. The antimicrobial activity of the silver-enriched beetroot–gelatin samples produced using gelatin solutions with different pHs is shown in Table 2. It was found that the best antimicrobial properties were present in the experimental samples produced using the gelatin-based solutions that had pH = 3 and pH = 4 in the presence of *E. coli* as well as pH = 7 and pH = 8 in the presence of *C. parapsilosis* and *A. fumigatus.* This indicated that, by varying the samples' pHs, it was possible to control the antimicrobial effect of the Ag containing samples on the microorganisms.

**Table 2.** Antimicrobial effectiveness of Ag containing samples produced using gelatin solutions with different pHs.

| Microorganism | pH | Control (Beetroot Juice) | BuAg10UV | BuAg20UV | BuAg30UV | BuAg40UV | BuAg50UV |
| --- | --- | --- | --- | --- | --- | --- | --- |
| | | | Zone Size, mm | | | | |
| *Escherichia coli* | 3 | 0 | 12.0 ± 0 | 13.5 ± 0.5 | 12.5 ± 0.5 | 12.5 ± 0.5 | 13.0 ± 0 |
| | 4 | 0 | 22.0 ± 0 | 17.5 ± 0.5 | 19.0 ± 0.5 | 22.5 ± 0.5 | 20.0 ± 0 |
| | 7 | 0 | 0 | 7.6 ± 0.3 | 0 | 0 | 0 |
| | 8 | 0 | 0 | 0 | 0 | 0 | 0 |

**Table 2.** *Cont.*

| Microorganism | pH | Control (Beetroot Juice) | BuAg10UV | BuAg20UV | BuAg30UV | BuAg40UV | BuAg50UV |
|---|---|---|---|---|---|---|---|
| | | | Zone Size, mm | | | | |
| *Pseudomonas aeruginosa* | 3 | 0 | 0 | 0 | 0 | 0 | 0 |
| | 4 | 0 | $5.0 \pm 0$ | $5.0 \pm 0$ | $5.0 \pm 0$ | $5.0 \pm 0$ | $5.0 \pm 0$ |
| | 7 | 0 | $12.0 \pm 0$ | $13.5 \pm 0.5$ | $15.0 \pm 0$ | $15.5 \pm 0.5$ | $12.0 \pm 0$ |
| | 8 | 0 | $13.0 \pm 0$ | $8.5 \pm 0.5$ | $10.5 \pm 0.5$ | $15 \pm 0.5$ | $13.0 \pm 0$ |
| *Candida parapsilosis* | 3 | 0 | 0 | 0 | 0 | 0 | 0 |
| | 4 | 0 | 0 | 0 | 0 | 0 | 0 |
| | 7 | 0 | $3.0 \pm 0$ | $7.0 \pm 0$ | $7.0 \pm 0$ | $6.0 \pm 0$ | $5.0 \pm 0$ |
| | 8 | 0 | $5.0 \pm 0$ | $7.5 \pm 0.5$ | $16.5 \pm 0.5$ | $8.0 \pm 0$ | $7.0 \pm 0$ |
| *Yarrowia lipolytica* | 3 | 0 | 0 | 0 | 0 | 0 | 0 |
| | 4 | 0 | 0 | 0 | 0 | 0 | 0 |
| | 7 | 0 | 0 | 0 | 0 | 0 | 0 |
| | 8 | 0 | 0 | 0 | $8.5 \pm 0.5$ | 0 | 0 |
| *Aspergillus terreus* | 3 | 0 | 0 | 0 | 0 | 0 | 0 |
| | 4 | 0 | 0 | 0 | 0 | 0 | 0 |
| | 7 | 0 | 0 | 0 | $10.5 \pm 0.5$ | 0 | 0 |
| | 8 | 0 | 0 | 0 | $12.0 \pm 0$ | 0 | 0 |
| *Aspergillus fumigatus* | 3 | 0 | 0 | 0 | 0 | 0 | 0 |
| | 4 | 0 | 0 | 0 | 0 | 0 | 0 |
| | 7 | 0 | $11.0 \pm 0$ | $8.5 \pm 0.5$ | $8.0 \pm 0$ | $10.0 \pm 0$ | $7.0 \pm 0$ |
| | 8 | 0 | $12.0 \pm 0$ | $12.5 \pm 0.5$ | $14.5 \pm 0.5$ | $11.5 \pm 0.5$ | $12.0 \pm 0$ |
| *Penicillium expansum* | 3 | 0 | 0 | 0 | 0 | 0 | 0 |
| | 4 | 0 | 0 | 0 | 0 | 0 | 0 |
| | 7 | 0 | 0 | 0 | $9.0 \pm 0$ | 0 | 0 |
| | 8 | 0 | 0 | $15.0 \pm 0$ | 0 | 0 | 0 |

## 4. Conclusions

The green synthesis method of silver NPs using beetroot juice as reducing and capping agents has been proposed and implemented for the production of silver-enriched antimicrobial recyclable free-standing films for their possible application in food packaging. In order to obtain free-standing films, a mixture of fresh beetroot juice and gelatin–water solution with a certain amount of admixed $AgNO_3$ was prepared and exposed to UV light.

It was shown that, at low concentrations of 1M $AgNO_3$ (10–30 μL), clusters of small Ag seeds were produced. At higher silver nitrate concentrations, a number of differently sized Ag nanoparticles were produced, and a particle growth tendency was observed. The produced particles were characterized through XRD as cubic phase metallic Ag particles. Due to the relatively low concentrations of $AgNO_3$ in the initial beetroot–gelatin mixture, the number of the produced particles was modest, but they were well dispersed in the whole film volume, thus indicating the antimicrobial effectiveness of the produced films. It was shown that the antimicrobial effectiveness was dependent on the pH of the solution. The best antimicrobial effectiveness was achieved at pH = 3 and pH = 4 in the case of *E. coli* as well as at pH = 7 and pH = 8 in the case of *C. parapsilosis* and *A. fumigatus*.

**Author Contributions:** Conceptualization, J.P. and D.A.; methodology, J.P.; investigation, J.P., A.P., D.A. and A.V.; writing—original draft preparation, J.P.; writing—review and editing, J.P. and D.A. All authors have read and agreed to the published version of the manuscript.

**Funding:** This research received no external funding.

**Institutional Review Board Statement:** Not applicable.

**Informed Consent Statement:** Not applicable.

**Data Availability Statement:** The data presented in this study are available on request from the corresponding author.

**Conflicts of Interest:** The authors declare no conflict of interest.

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
