# Peer review of "Investigation of the Antimicrobial Properties of Beetroot–Gelatin Films Containing Silver Particles Obtained via Green Synthesis"

_applsci, doi:10.3390/app13031926_

Round 1

Reviewer 1 Report

The article is devoted to obtaining and characterizing the antimicrobial properties of silver nanoparticles in a film of gelatin and beets. The biggest problem of the article is just the same lack of physical and chemical characteristics. The authors were unable to prove the presence of silver nanoparticles, their size and distribution in the gelatin film. The article needs serious revision.

1.       Line 109: “Weighed amount of silver colloid were casted onto plastic Petri dishes and then dried 108 in open air in the dark box at the room temperature (20±20C) for 72 hours”. Have you really used such a wide range of temperatures?

2.       Line 107: What is the reason for obtaining films? Not solution or others

3.       Line 142: “Evaluation of the pH effect on the antimicrobial properties of the gelatine films with silver nano- 142 particle” It is not very clear at what stage the system's pH was changed? Were blank tests performed with different pHs, but without silver nanoparticles? Could microorganisms die simply in acidic solution, without the influence of nanoparticles?

4.       Line 197: “The XRD spectra are presented in Fig. 6b”. Where is fig 6b? Please add

5.       Line 199-201: “. The peaks at 18.61, 21, 45, 30, 45 2θ◦ can be assigned to (111), (200) and (220) planes of cubic AgCl (JCPDS file No. 00-031-01238), 200 whereas the peaks at 21.15, 29.24 2θ◦ can be assigned to (111), (200) and (220) planes of cubic 201 Ag (JCPDS file No. 00-004-0783)”. Why is one peak for chloride given with an accuracy of hundredths, and the rest are only integer values? Why are you giving five peaks for three Miller indices? Why, on the contrary, do two peaks for silver correspond to three Miller indices?

6.       Line 203: “Especially, the strong peak at 21.47 2θ◦ was coincident with the 2.77 Å in AgCl [200] 203 layer from HR-TEM analysis [24-25].” how can it be that the reflections coincide at different angles?

7.       Line 204-205: “In addition, there is no peaks of metal Ag in XRD patterns, the possible reason is that the content of these nanoparticles is very small and is cannot be showed in the XRD image.” Above you say that peaks for metallic silver have been found. How then to understand you?

8.       Line 218: “Ag nanoparticles in beetroot gelatine composite are hardly identified by SEM”. Are you sure that you actually have silver nanoparticles? You declare their presence in the abstract and in the title of the article, but do not prove it in any way

9.       Line 219: “charge effect on nonconductive samples is the reason.” write the conditions for the SEM. Did you use spray?

10.   Line 226: “The Ag nanoparticles size according the SEM measurements are 226 very small (1- 2 nm).” On the one hand, you do not provide clear SEM photographs to prove the size of the nanoparticles, and on the other hand, you state above that the resolution was low. Please provide SEMs in the article to prove the size. Plus, where did you get the idea that it is metallic silver that has such a small size, without a shell of silver chloride?

11.   Line 282: “According SEM results Ag nanoparticle well dispersed and non- aggregated in the film in gelatine –beetroot juice films volume. Cubic phase of metallic Ag coexisting with the cubic phase of AgCl are dispersed in the crystalline gelatine beetroot juice films” The authors do not support these conclusions. They need to be reviewed.

Author Response

Dear Reviewer,

Thank you for reviewing our paper. We appreciate your comments and suggestions for improvements. The answers to your questions are provided below

Reviewer 2 Report

This manuscript reports the preparation of plasmonic silver nanoparticles via photoreduction in gelatin. The silver nitrate ions were reduced by the natural pigments found in Beetroot which was the source used. The silver nanoparticles in gelatine – beetroot nanocomposite film was evaluated for antimicrobial activities. 

1.   On page 3, line 109, it says “(20±20C)”. I’m assuming that the authors are trying to indicate that the temperature standard deviation is 2 oC. if so, the statement should be “(20 ± 2 oC)”. 

2.   The authors mentions that the peak in Figure 6 around 20 – 20.90 2θobelongs to gelatine. The Figure isn’t very clear as to the presence of this peak. The authors should include the XRD pattern for gelatine so that it can be compared to the silver nanoparticles in gelatine – beetroot nanocomposite film. Figure 6 is incorrectly label; it should say “X-Ray Diffraction pattern of….”. 

3.   Also, the authors mentioned “Fig. 6b” on page 6 but there’s no evidence of Figure 6b in the manuscript. 

4.   The authors mentioned that there’s a peak at 8.3 2θo, but didn’tstate what is attributing to this peak. 

Author Response

(The authors gave the same response as above.)

Reviewer 3 Report

This is a very interesting paper and method. The authors do a nice job of introducing the topic and need for this method. 

I have some concerns about the methods/conclusions. One major concern that I have is that the LSPR shift with silver ion concentration was mentioned and demonstrated in Figure 5, however, it was not explained. 

Detailed comments:

Line 19 – Why is it “relatively” reproducible?

Lines 93-96 – This seems to be lacking some detail. For example, how was the grated beetroot filtered? The silver nitrate needs more details as well – solid or solution? Purity?

Line 100 – is this supposed to be micromolar? How was this beetroot concentration determined?

Line 101 – It is unclear if these solutions were combined before irradiation. It states the final solutions, which implies they were separate, which also matches the figure, but in line 100 it implies they were combined.

Line 109 – 20 +/- 20C seems like a very large temperature range. Is this correct?

157 – how was this sugar content determined?

Figure 3 – why do the absorbance wavelengths on the UV-Vis spectra shift with changing concentration? Is this figure just pure beetroot juice, or was something added?

Line 165 – details on the UV-Vis irradiation? Was this just from scanning?

Line 179 – Why did the LSPR peak shift with changing concentration? Did the size of the resulting nanoparticles change, resulting in the LSPR shift? I would expect larger nanoparticles with higher concentration, so this shift seems a bit counterintuitive. Additional discussion/characterization is warranted.

Figure 6 – I believe the caption is incorrect

Line 218 – 227 – I am not understanding this paragraph. It seems as though the authors state that they were unable to get high enough resolution to image the AgNPs, however, the last line states that the AgNP size was 1-2 nm according to SEM. Can this be clarified?

Line 225 – “. The conductive polymer dissolved the investigated samples and created new nanocomposite.” – Please clarify this

Line 278 – “The absorbance intensity 278 and position of LSPR of silver nanoparticles in the films depending on primary silver ion concentrations in the beetroot and gelatine with diluted beetroot’s juice solutions.” – This needs additional justification/investigation if it is a major conclusion of the paper.

Author Response

(The authors gave the same response as above.)

Round 2

Reviewer 1 Report

I thank the authors for their answers. However, I still advise the authors to bring significant evidence of the presence of silver nanoparticles.

1. On the given XRD patterns, the peaks indicated for nanoparticles do not exceed the noise intensity and do not prove the presence of nanoparticles.

2. SEM micrograph data cannot be analyzed due to their low quality.

I advise the authors to repeat the syntheses and provide qualitative data in the manuscript.

Author Response

Dear Reviewer, 

Thank you for reviewing our paper. We appreciate your comments and suggestions for improvements. The answers to your questions are provided below.

Reviewer 2 Report

1. On page 3, lines 97 & 98, it says that "The resulting mixture was filtered through 8 - 12 m of black band filter". Is 8 - 12 m correct or is the size of the filter much smaller? 

2. In some instances in the manuscript, the number 3 in AgNO3 was not subscripted. It would be great to have consistency with the reporting. 

3. On page 9, line 284, please include "Fig.8d" at the end of the sentence. Also, for the caption in Figure 8, figures 8c and 8d were not mentioned. 

Author Response

(The authors gave the same response as above.)
